# Investigation of *Astyanax mexicanus* (Characiformes, Characidae) chromosome 1 structure reveals unmapped sequences and suggests conserved evolution

Maelin Silva[1,2], Duílio Mazzoni Zerbinato Andrade Silva[3]*, Jonathan Pena Castro[2], Alex I. Makunin[4‡], Felipe Faix Barby[5‡], Edivaldo Herculano Correa de Oliveira[6‡], Thomas Liehr[7‡], Marcelo Bello Cioffi[5‡], Fábio Porto-Foresti[8‡], Fausto Foresti[9‡], Roberto Ferreira Artoni[2,5]

1 Instituto Latino-Americano de Ciências da Vida, Universidade Federal da Integração Latino-Americana, Foz do Iguaçu, Paraná, Brazil, 2 Departamento de Biologia Estrutural, Molecular e Genética, Programa de Pós-Graduação em Biologia Evolutiva, Universidade Estadual de Ponta Grossa, Ponta Grossa, Paraná, Brazil, 3 Cell and Developmental Biology Center, National Heart, Lung, and Blood Institute, National Institutes of Health, Bethesda, Maryland, United States of America, 4 Wellcome Sanger Institute, Hinxton, Cambridgeshire, United Kingdom, 5 Programa de Pós-Graduação em Genética Evolutiva e Biologia Molecular, Universidade Federal de São Carlos, São Carlos, São Paulo, Brazil, 6 Instituto de Ciências Exatas e Naturais, Universidade Federal do Pará e Laboratório de Citogenômica e Mutagênese Ambiental, Seção de Meio Ambiente, Instituto Evandro Chagas, Belém, Pará, Brazil, 7 Institute of Human Genetics, Jena University Hospital, Friedrich Schiller University, Jena, Germany, 8 Departamento de Ciências Biológicas, Faculdade de Ciências, Universidade Estadual Paulista - UNESP, Bauru, São Paulo, Brazil, 9 Departamento de Biologia Estrutural e Funcional, Instituto de Biociências de Botucatu, Universidade Estadual Paulista - UNESP, Bauru, São Paulo, Brazil

☯ These authors contributed equally to this work.
‡ AIM, FFB, EHCO, TL, MBC, FPF and FF also contributed equally to this work.
* duilio.silva@nih.gov

## Abstract

Natural selection in the cave habitat has resulted in unique phenotypic traits (including pigmentation loss and ocular degeneration) in the Mexican tetra *Astyanax mexicanus*, considered a model species for evolutionary research. *A. mexicanus* has a karyotype of 2n = 50 chromosomes, and long-read sequencing and quantitative trait linkage maps (QTLs) have completely reconstructed the reference genome at the chromosomal level. In the current work, we performed whole chromosome isolation by microdissection and total amplification using DOP-PCR and Whole Chromosome Painting (WCP), followed by sequencing on the Illumina NextSeq platform, to investigate the microstructure of the large and conserved metacentric chromosome 1 of *A. mexicanus*. The sequences aligned to linkage block 3 of the reference genome, as determined by processing the reads with the DOPseq pipeline and characterizing the satellites with the TAREAN program. In addition, part of the sequences was anchored in linkage blocks that have not yet been assigned to the chromosomes. Furthermore, fluorescence in situ hybridization using WCP 1 carried out in other nearby species revealed a high degree of chromosome conservation, which allows us to hypothesize a common origin of this element. The physical mapping of the repetitive marker sequences provided a micro- and macrostructural overview and confirmed their position in

**Data Availability Statement:** All sequences are publicly deposited in the NCBI read archive, with accession number SUB12147369.

**Funding:** This study received financial support from the Conselho Nacional de Desenvolvimento Científico e Tecnológico (CNPq) (grant 407187/2016-2). The author M.S. received a scholarship from the Conselho Nacional de Desenvolvimento Científico e Tecnológico (CNPq) (grant 160155/2018-5). The author E. H. C. de O. received a scholarship from the Conselho Nacional de Desenvolvimento Científico e Tecnológico (CNPq) (grant 307382/2019-2). https://www.gov.br/cnpq/pt-br. The author D. M. Z. A. S. received a scholarship from the Fundação de Amparo à Pesquisa do Estado de São Paulo (grant 2017/22447-8). https://fapesp.br/. Funders did not play any role in the study design, data collection and analysis, decision to publish, or preparation of the manuscript.

**Competing interests:** The authors have declared that no competing interests exist.

chromosome pair 1. These sequences can serve as comparative tools for understanding the evolution and organization of this chromosome in other species of the family in future studies.

## Introduction

*Astyanax* (Baird and Girar, 1854) species are predominantly found in South America, inhabiting diverse environments such as mountainous regions, springs, and lagoon areas. They are distributed throughout the Neotropical region, ranging from the southeastern United States to the central region of Argentina [1–3]. With over 127 recognized species, *Astyanax* is one of the most speciose genera in the Characiformes order [4]. A recent phylogenetically based reevaluation of the genus *Astyanax* resulted in the reassignment of some species that were formerly placed in the genus to the genera *Psalidodon*, *Andromakhe*, *Deuterodon*, and *Makunaima* [5]. The extant *Astyanax* species possess 2n = 50 chromosomes, with 5S rDNA limited to submetacentric chromosomes and few As51 satellite DNA sites [6–9].

The blind tetra *Astyanax mexicanus* (De Filippi, 1853) is an iconic species inhabiting caves, specifically distributed in Central and North America. This species has developed peculiar phenotypic characteristics and is considered a model organism for studies on cave adaptation [10]. The morphology of *A. mexicanus* has evolved through natural selection, adapting specific traits such as loss of pigmentation, complete absence of eyes due to ocular degeneration, and enhancement of tactile and sensory organs as compensatory mechanisms [11]. Recently, a high-resolution reference genome was assembled at the chromosomal level for *A. mexicanus*, which has emerged as a model for the evolutionary analysis of complex traits associated with troglomorphic features [12, 13].

*A. mexicanus* has a diploid number of 2n = 50 chromosomes with a karyotype composed of 8m+18sm+12st+12a, in addition to the presence of B microchromosomes in some individuals [14]. Remarkably, its first chromosomal pair is a synapomorphy conserved in all *Astyanax* species, including those previously assigned to the genus [5, 15–17]. This chromosome is easily identified in the karyotype as the largest metacentric chromosome, distinguished from other chromosome pairs by its symmetric morphology between the short and long arms. It is almost entirely euchromatic, with heterochromatic blocks distributed on both arms in the telomeric region [16, 18].

Other Characidae species, including those previously assigned to *Astyanax*, also exhibit chromosome 1 in their karyotype as a meta-submetacentric macrochromosome, maintaining its morphology consistently over time [19]. This characteristic suggests that this linkage group might also be preserved in genetic mapping. The conservation of chromosome 1 may date back to a period before the estimated 3.3 Mya final closure of the isthmus of Panama, which led to the separation between *A. mexicanus* and other *Astyanax* species and related groups in the Neotropical region [1]. Therefore, a comprehensive sequencing study of *A. mexicanus's* linkage group of chromosome pair 1 and the obtainment of specific probes for this chromosome, constitute important tools for the taxonomy and comprehension of these fish's diversity. These methods can also yield important insights into the biological roles played by the genes connected to this linkage group.

Consequently, the main objective of this study was to sequence chromosome 1 of *A. mexicanus* using Next Generation Sequencing (NGS), generate an efficient probe for fluorescence in situ hybridization (FISH) mapping, and isolate satellite DNAs (satDNAs) present in this

chromosome. These analyses aim to provide a detailed understanding of the microstructure and repetitive sequences of this chromosome, contributing to the taxonomy and understanding of the diversity of these species.

## Material and methods

### Characterization of the study subject

We used 5 animals for each species: *Astyanax mexicanus*, purchased from an aquarium store, identified and deposited at the Museu de História Natural do Capão da Imbuia (MHNCI voucher number 13035), and other species collected in the wild: *A. altiparanae*, from the upper Tibagi river (26˚ 17'13.98 S and 51˚ 17'54.18 W), MHNCI 11460; *Psalidodon bifasciatus*, from the middle Iguaçu river (26˚ 15' 1.11″ S and 51˚ 6' 10.67″ W), identified and deposited in the Ichthyological Collection of the Núcleo de Pesquisas de Limnologia, Ictiologia e Aquicultura—NUPÉLIA (NUP no. 16898), at the Universidade Estadual de Maringá –UEM, and *P. scabripinnis*, from Fazenda Lavrinha, a stream in the Paraiba do Sul river basin (22˚43′09,6′S and 45˚25′38,5′′W), (NUP no. 17482).

Specimens were anesthetized with 0.1% benzocaine and sacrificed. All procedures followed ethical guidelines for animal experimentation detailed by the Ethics Committee of the State University of Ponta Grossa (Protocol SEI UEPG number 23.000061551–4 and UEPG Ethics Committee on the Use of Animals (CEUA) number 0769342/2021). The license to collect biological material was granted by Instituto Brasileiro do Meio Ambiente e dos Recursos Naturais Renováveis (IBAMA) under number 15115–1.

### Cell culture and obtaining mitotic metaphase chromosomes

For the FISH experiments, we used metaphase chromosomes of *A. mexicanus* from cell cultures, obtained following the protocol for rapid culture of teleost fish fibroblasts [20]. Briefly, muscular tissue, dorsal and pectoral fin tissues were disaggregated in type IV collagenase and cultured at 37 ˚C in Dulbecco's Modified Eagle's Medium high glucose (GIBCO™, Grand Island, NY, USA), supplemented with 20% fetal bovine serum (GIBCO™, USA), penicillin (100 units/ml), and streptomycin (100 mg/ml). The cells were cultured until ~80–90% confluence. For subsequent passages, the cells were removed by adding Trypsin-EDTA (GIBCO™, USA), and up to five passages were performed. To confirm the diploid number and verify if there were no chromosomal alterations for each new culture, chromosomes were obtained using standard protocols: the cells were incubated for 3 h with colchicine, treated for 8 min in a hypotonic solution (0.075 M KCl), and fixed in methanol and acetic acid solution (3:1).

For the other species (*A. altiparanae*, *Psalidodon bifasciatus*, and *P. scabripinnis*), we employed the technique of cell suspension, commonly used for studying chromosomes in fishes, according to [21].

### Chromosome 1 microdissection, validation, and labeling

*A. mexicanus* mitotic chromosomes were prepared on a coverslip and stained in Giemsa 5% (v/v) for 5 min and washed in distilled water. Chromosomal microdissections were performed using an inverted microscope (Zeiss) equipped with a mechanical microdissector (Narishige) following the protocol described by [22]. Briefly, 15 copies of the first pair were microdissected with capillary needles with tips measuring approximately 0.7 µm. After microdissecting the chromosome, the tip of the needle was broken inside a microtube where amplification was carried out by Degenerate Oligonucleotide-Primed—Polymerase Chain Reaction (DOP-PCR). Ten microdissected chromosomes were added for each reaction. The microdissected

chromosomes were previously treated with the enzyme Topoisomerase I. We used 0.5 μl of Topo buffer and 4 μl of dH$_2$O for each tube and incubated for 10 min at 100˚C. After which the tubes were immediately subjected to 4˚C for heat shock. Subsequently, 0.1 μl of Topoisomerase I was added and incubated for 1 h at 37˚C, and 15 min at 65˚C.

DOP-PCR reactions followed the general procedure described by [23], with some modifications. The procedure uses 20 μl of PCR reaction mix (5.0μM DOP primer 5' `CCGACTCGAGN NNNNNATGTGG` 3'; 200 μM dNTPs; 1.6 U/μl sequenase V2 U (T7 DNA polymerase, USB, Cleveland, OH, USA); 0.6X Sequenase Buffer; 1.0 U/μl Amplitaq$^{®}$ 360 DNA Polymerase AB (Applied Biosystems™); 2.50 mM MgCl$_2$; 0.10μl Amplitaq buffer). The PCR program consisted of denaturation at 92˚C for 5 min, followed by 8 cycles of low stringency (92˚C for 1 min, 30˚C for 2 min and 20 s, and 56˚C for 2 min and 20 s), followed by 32 cycles of high stringency (92˚C for 1 min, 56˚C for 1 min, 72˚C for 2 min) finishing the reaction at 72˚C for 10 min.). The products were checked on 1% agarose gel. 1μl of the first PCR product was used for a new amplification reaction in a final volume of 50μl (1x Amplitaq buffer; 0.2mM dNTPs; 1.0μM DOP primer; 2.5mM MgCl2; 0.1 U/μl Amplitaq), according to the following conditions: an initial denaturation at 92˚C for 3 min, followed by 30 cycles at 91˚C for 1 min, 56˚C for 1 min and 72˚C for 2 min, finishing at 72˚C for 5 min. The PCR product was rechecked on a 1% agarose gel.

To confirm if our microdissection covered the entire length of chromosome 1, we amplified sequences from three single-copy genes: two located near the ends of the chromosome and one near the centromere. To achieve this, we identified these genes in the species' assembled reference genome (GenBank GCF_000372685.2) and designed specific primers for each one of them: *Nyap1*, *Mllt10*, and *Pcdb* (see S1 Table).

The chromosome 1 probe was labeled directly by nick translation with Spectrum Orange-dUTP Reagent Kit (Abbott Molecular Inc., Abbott Park, IL) or indirectly with Dig Nick Translation Mix (Sigma-Aldrich) according to the manufacturer's instructions. The product of this reaction was precipitated with potassium acetate and ethanol for 30 min at -80˚C. Subsequently, the material was centrifuged for 25 min at 14,000 rpm, and the supernatant was discarded, leaving the labeled DNA to dry completely in an oven at 37˚C.

## Sequencing of the microdissected chromosome 1 and *in silico* mapping

The library for sequencing the microdissected chromosome 1 of *A. mexicanus* amplified by DOP-PCR was prepared with the Nextera XT Sample Preparation Kit (Illumina), which includes random fragmentation of the template DNA. The sample with barcode N7 (N701 – TAAGGCGA) and S5 (S503 –AGAGGATA) was sequenced on an Illumina NextSeq and NextSeq 500 MID Output kit (300 cycles) of Illumina$^{®}$ next-generation sequencing (paired-end 2x151 bp). All sequences are publicly deposited in the NCBI read archive, with accession number SUB12147369.

The chromosome 1 reads were mapped onto the *A. mexicanus* genome (Genbank GCF_000372685.2) using the DOPseq v2.1.0 pipeline with standard parameters [24, 25] (https://github.com/lca-imcb/dopseq). This pipeline was developed by [24] to investigate specific regions of isolated chromosomes.

## Prospecting for chromosome 1 satellite DNAs and transposable elements

To prospect for satDNAs in the *A. mexicanus* genome, we used the reads from the microdissected chromosome 1 and genomic reads available for *A. mexicanus* in the Genbank (SRA: SRS2765935). We aimed to identify satDNAs exclusive for chromosome 1 to better understand its repetitive DNAs microstructure. The reads were processed as follows. After quality and

adapter trimming using Trimmomatic v0.33 [26], we performed a high-throughput analysis of satDNAs within the *A. mexicanus* genome using the TAREAN software [27]. To get the full catalog of SatDNAs we used the satMiner script [28] as performed in [29]. To detect satDNAs present on chromosome 1, we used the following strategy. After obtaining two catalogs of satDNAs: from the whole genome reads and the chromosome 1 reads, we searched for those present only in the catalog of chromosome 1, as possible candidates to be exclusive to this chromosome. Furthermore, we searched for unique satellites from the whole genome library as candidates for being absent on chromosome 1 as controls of our FISH experiments. From this analysis, we selected four satDNAs present on chromosome 1 (SatA_mex, SatB_mex, SatC_mex and SatD_mex) and two absent in this chromosome (SatE_mex and SatF_mex). Based on their monomer sequences we designed convergent primers (S2 Table) for the amplification of these satDNAs and the production of probes for FISH. The similarity with satDNAs described in the literature is in the S3 Table.

To identify transposable elements in chromosome 1, we submitted the reads generated from this chromosome to the RepeatExplorer2 [30] software, using a reference library for the *A. mexicanus* genome generated by [31] with the RepeatMasker 4.0.7 (http://www.repeatmasker.org) software.

## Amplification and production of satellite DNA probes

PCR amplification reactions were performed for each of the satellite DNAs, starting from the genomic DNA of the species with the following parameters: 1X PCR Buffer, 1.5mM of $MgCl_2$, 100uM of dNTP, 0.1 uM of Primer A + Primer B, 0.1U of Taq polymerase and 2-100ng of genomic DNA, in a total reaction of 10ul. The primers used for amplification are described in the S2 Table. The amplification cycles varied according to the parameters required for each satellite DNA, with the cycle described as follows: initial denaturation at 95˚C for 5 min followed by 30 cycles of 95˚C for 30 s, 56˚C to 60˚C for 15–30 s, and 72˚C for 30 s, and extension of 72˚C for 7 min.

The production of the probes followed with PCR reaction for labeling, with the following parameters: 31 μl of Milli-Q water, 5 μl of Taq polymerase enzyme buffer (10X), 5 μl of $MgCl_2$ (25mM), 1 μl of dATP, dCTP and dGTP and 0.7 μl of dTTP (2mM each), 0.6 μl of Digoxigenin-11-dUTP or Biotin-16-dUTP (Roche Applied Science), 1μl of each primer (10mM), 0.5 μl of Taq polymerase (5U/ μl) and 2 μl of template DNA, totaling a reaction of 50 μl. The cycle parameters were the same used in the amplification reactions.

## Obtaining the Cot-1 repetitive DNA blocker

To obtain the Cot-1 repetitive DNA blocker, we used a DOP-PCR reaction to amplify the liver genomic DNA of the samples extracted using the phenol-chloroform method [32]. The PCR program consisted of 8 cycles of low stringency (92˚C for 1 min, 30˚C for 1 min 30 s, and 72˚C for 2 min) followed by 25 cycles of high stringency (92˚C for 1 min, 62˚C for 1 min 30 s, and 72˚C for 2 min). The reaction product was reamplified following 30 cycles of 91˚C for 1 min, 56˚C for 1 min, and 70˚C for 2 min. After obtaining the repetitive blocking DNA, it was used for blocking in the FISH of the chromosome 1 probe, in a 3:1 ratio of blocker-probe.

## Fluorescence *in situ* hybridization

We performed FISH with the whole chromosome probe for chromosome 1 in *Astyanax mexicanus*, *A. altiparanae*, *Psalidodon bifasciatus*, and *P. scabripinnis* and satDNAs in *A. mexicanus* and in *P. scabripinnis* following the protocol described by [33] with adaptations: After pretreatment and initial denaturation of the chromosomal material, we applied 20 μl of

hybridization solution (containing 2.5 ng/μl probe, 7.5ng/μl Cot-1 DNA, 50% formamide, 2×SSC, and 10% dextran sulfate) to the slides. These were then kept at 37˚C for 48 h in a humidified chamber. Afterwards, the slides were washed with a 15% formamide solution in 2xSSC for 15 min at 42˚C, followed by a wash with 1xSSC at 60˚C for 15 min. The slides were then rinsed in Tween 0.5%/4xSSC for 5 min, in 5% NFDM/4xSSC buffer for 15 min, and again in Tween 0.5%/4xSSC twice, each for 5 min.

For the indirectly labeled chromosome 1, it was incubated with 30 μl of blocking solution containing Anti-Digoxigenin-AP, Fab fragments from sheep (Sigma-Aldrich) antibodies for 1 h in a dark, humidified chamber at 37˚C. The satellite DNA probes were treated similarly, but with either Anti-Digoxigenin-AP, Fab fragments from sheep (Sigma-Aldrich) or Streptavidin Alexa Fluor 488 conjugate (Thermo Fisher) antibodies. For the chromosome 1 probe directly labeled with Spectrum Orange-dUTP (Abbott Molecular Inc., Abbott Park, IL), this step was skipped.

After that, all the slides were then washed 3 times in Tween 0.5%/4xSSC, 5 min each, followed by dehydration in a series of 70%, 85%, and 100% ethanol at room temperature, 5 min in each bath. The chromosomes were then counterstained with DAPI (0.2 mg/ml) diluted in an anti-fading solution (Fluka) or with propidium iodide (50 μg/ ml). Chromosome preparations were then analyzed under a Zeiss AxioCam MRm epifluorescence microscope with ZEN pro-2011 software (Carl Zeiss).

## Results and discussion

Through chromosomal microdissection of chromosome pair 1 (Fig 1a) and subsequent amplification by Degenerate-Oligonucleotide-Primed—Polymerase Chain Reaction (DOP-PCR), we were able to isolate and amplify the total DNA of this chromosome specifically. The isolated product was then used as a probe in chromosomal painting, which resulted in a strong labelling of the entire chromosome pair 1 in its own species (*A. mexicanus*, Fig 1b), as well as in *A. altiparanae* (Fig 1c), *Psalidodon bifasciatus* (Fig 1d), and *P. scabripinnis* (Fig 1e).

In addition, small fragments of the probe can be observed in other chromosomes, probably due to the presence of common repetitive DNAs that were not completely suppressed by the Cot-1 DNA used for blocking (Fig 1B–1E).

The NGS reads of chromosome 1 were mostly anchored in the largest linkage block of the *A. mexicanus* karyotype (Genbank GCF_000372685.2) represented by chromosome 3 (Genbank NC_035899.1) in the reference genome (Fig 1F). A portion of the reads were anchored in scaffolds not yet assigned to chromosomes (S4 Table).

Although repeated sequences are preferentially amplified throughout DOP-PCR procedure for probe preparation and amplification after microdissection procedures, fragments of 400 bp of the *Nyap1*, *Mllt10* and *Pcdb* genes were successfully amplified via PCR from the microdissected and DOP-PCR amplified material of chromosome 1. This demonstrates that the isolated material keeps a good representation of low-repeated elements and/or single copy genes (see material and methods).

Physical mapping by FISH in *A. mexicanus* of satDNAs selected for being present or absent (negative control) on chromosome 1 revealed a block distribution of all satDNAs with none being exclusively distributed on pair 1 (Fig 2A–2E). However, they showed a particular distribution pattern on this chromosome. The SatA_mex (574bp) is located in the distal portion of both arms of chromosome 1; in addition, this sequence is distributed across several complement chromosomes in the heterochromatic regions of centromeres and telomeres (Fig 2A). SatB_mex (584bp) is located pericentromerically in the two arms of chromosome 1, and nine more pairs of complement A in pericentromeric regions (Fig 2B). Meanwhile, SatC_mex

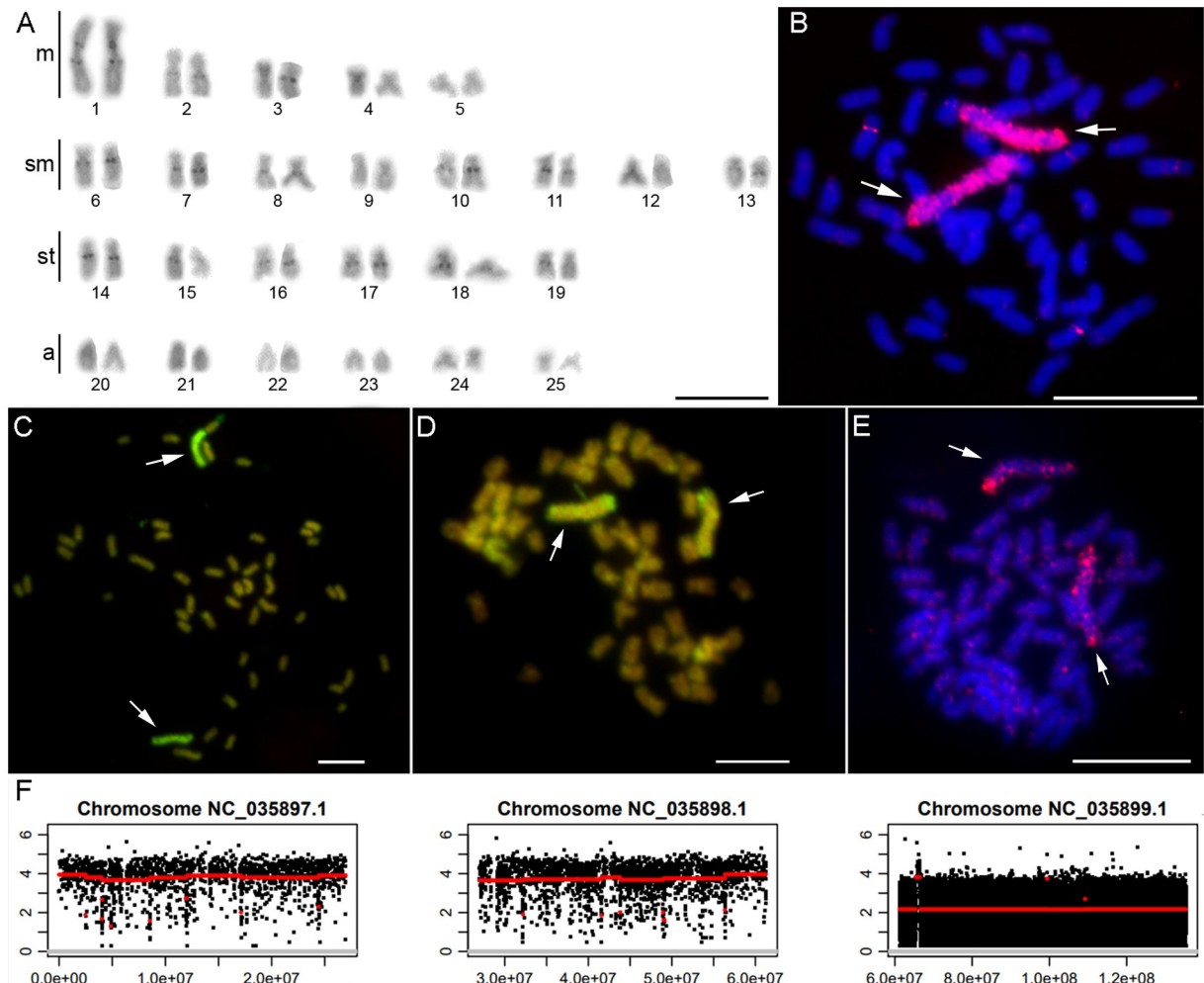

**Fig 1. Karyotype, mapping of the probe and anchored reads of chromosome 1 of *Astyanax mexicanus*.** A) Conventional Giemsa staining revealed a karyotype for *A. mexicanus* as 2n = 50. B) FISH-mapping of the probe isolated by microdissection of chromosome 1 in *A. mexicanus*, in *A. altiparanae* (C); in *Psalidodon bifasciatus* (D); and *P. scabripinnis* (E). In B and E the probe is marked with digoxigenin—rhodamine (red), the chromosomes are counter-stained by DAPI (blue), and in C and D the probe is marked with Spectrum Orange-dUTP and counter-stained with propidium iodide. The arrows indicate the chromosome 1 pair. Bar: 10 µm.; F) Reads anchored in the first three chromosomes of the karyotype, represented in the reference genome.

(178bp) is located only in a band on chromosome 1 between SatA_mex and SatD_mex, also close to the centromeres, and distributed in the centromeric region of six more pairs (Fig 2C). The satellite with the widest distribution on chromosome 1 was SatD_mex (54bp), which was present in four blocks arranged in two equidistant bands on each chromosomal arm. Additionally, small blocks of SatD_mex were observed in most of the A complement chromosomes (Fig 2D).

The sequence used as a negative control, SatE_mex, was found in three pairs of the standard complement but not on chromosome 1, confirming the specificity of the isolated chromosome 1 DNA (Fig 2E). Conversely, the SatF_mex sequence did not produce any visible signal via FISH. The signals and schematic map of all the Satellite DNAs mapped on chromosome 1 are illustrated in Fig 2F and 2G.

The physical mapping of the satDNAs_mex in *P. scabripinnis* by FISH (Fig 3) revealed similarities with the results obtained in *A. mexicanus* for SatB_mex and SatD_mex (Fig 2B and

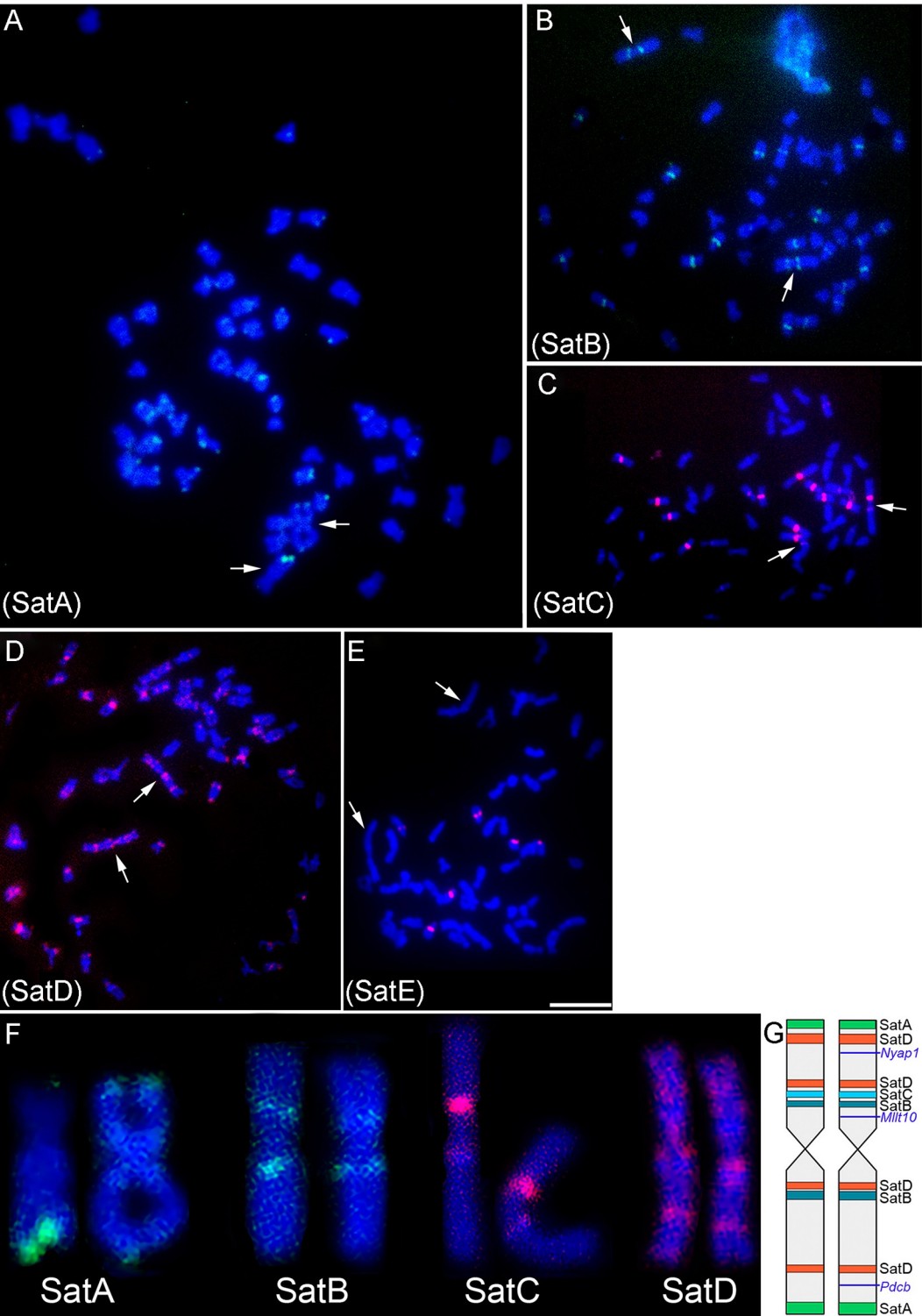

**Fig 2. Mapping of satellite DNAs markers for chromosome 1 of *Astyanax mexicanus*.** A) SatA_mex (574bp), B) SatB_mex (584 bp), C) SatC_mex (178bp) and D) SatD_mex (54bp). The satellites SatE_mex (229 bp) and SatF_mex (152bp), revealed signals outside of chromosome 1, used as a negative control. The sequences were marked with Streptavidin (green) or Digoxigenin—rhodamine (red). The arrows indicate chromosome 1 pair. Bar: 10 μm. F) Chromosome 1 and its satellite DNA distribution. G) Idiogram of all satellite DNAs mapped on chromosome 1 (left homologue) and with the satellite DNAs and the exclusive monogenic amplified sequences mapped (right homologue).

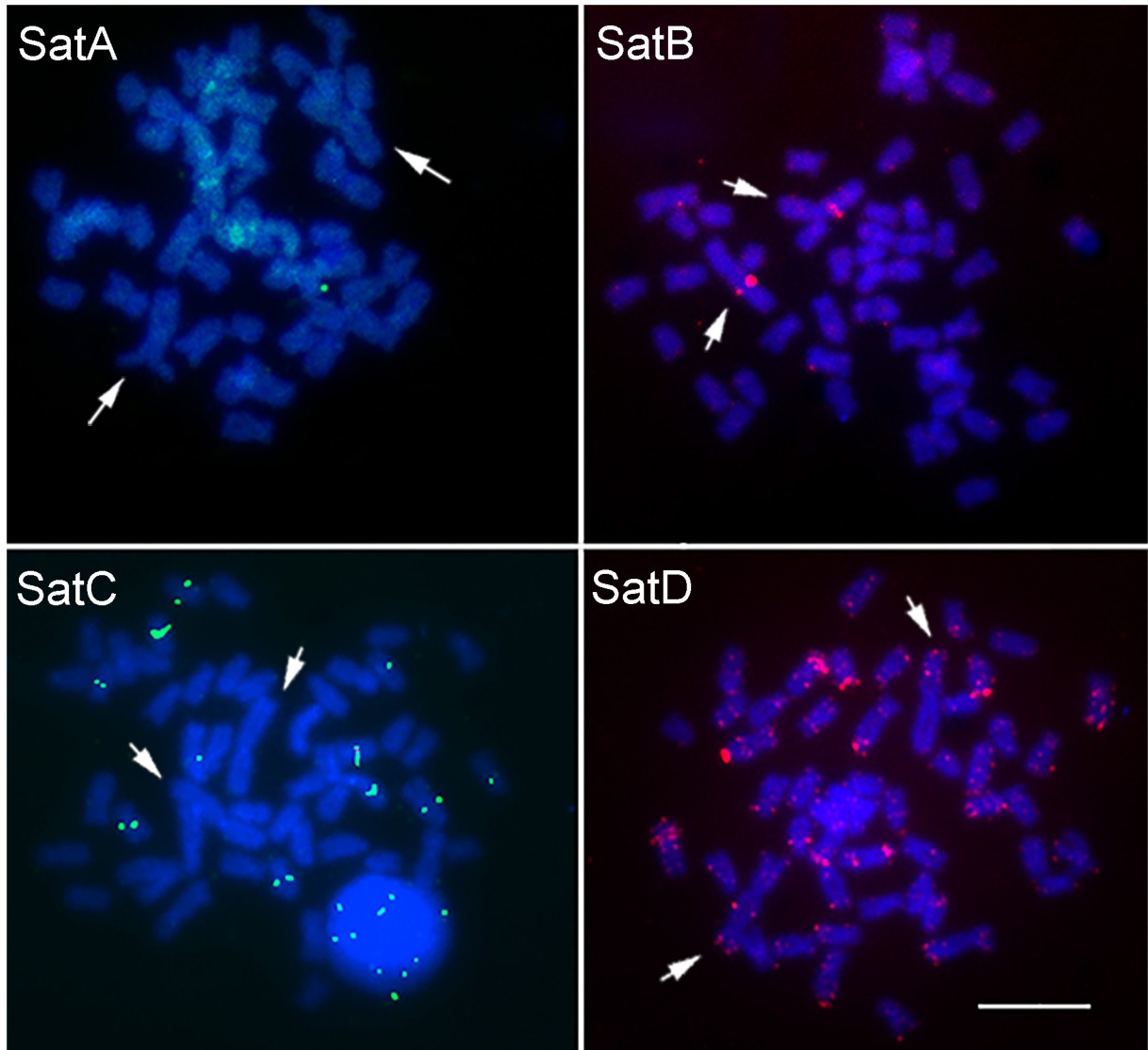

**Fig 3. Signals of satellite DNAs probes for chromosome 1 of *A. mexicanus* in *Psalidodon scabripinnis*.** The sequences were marked with Streptavidin (green) or Digoxigenin—rhodamine (red). The arrows indicate the chromosome pair 1. Bar: 10 μm.

2D). However, for SatA_mex and SatC_mex in *P. scabripinnis*, no signals were observed on chromosome 1 (as shown in Fig 3). Additionally, SatE_mex and SatF_mex did not produce any visible signals by FISH.

The chromosome microdissection technique is useful for isolating specific chromosomes or chromosomal regions of interest, which can be amplified later for different purposes. Labeled DNA sequences can be obtained to perform FISH with WCP probes specific for particular chromosomal regions [34]. Through high-throughput sequencing on NGS platforms, the detailed content of the isolated region is assessable, making it a profitable and productive method for acquiring specific genomic content [35, 36].

The presence of synapomorphic chromosome pair 1 in the genus *Astyanax* allows us to hypothesize a common origin for this element. Furthermore, its presence in other species of the Characidae family [1, 19] permits us to investigate whether this origin is shared with other

members of the family and how long ago this chromosome originated. To investigate these hypotheses, we produced a reliable probe of the chromosome pair 1 of *A. mexicanus*. Our results with related nearby species (*A. altiparanae* and *Psalidodon bifasciatus* and *P. scabripinnis*) demonstrated the homology and conservation of this chromosome in its macrostructure, showing that the origin of this chromosome can date back to at least 3.3 Mya, when these *Astyanax* and *Psalidodon* species were separated. The satellites isolated and mapped in *A. mexicanus* and *P. scabripinnis* demonstrate dynamic rearrangements despite the maintenance of the general morphology of this chromosome.

The combination of cytogenetic methods and NGS to explore the repetitive portion of chromosomes employed here has also been successfully applied in other studies to both supernumerary and sex chromosomes. In a study combining both methods, [36] identified 17 different satDNAs, of which 12 appear accumulated in centromeric and telomeric regions of the chromosomes in the three species of geckos. The authors hypothesize that the great stability of the fully acrocentric karyotype in the three species can be explained by the presence of specific satDNAs at the centromeric regions that are strong meiotic drivers. In another way [35] were successful in identifying the microstructure of the B chromosome of the lizard *Anolis carolinensis*, performing isolation by flow-sorting, followed by amplification by DOP-PCR and sequencing on the Illumina MiSeq Sequencing platform. Specific DNA sequencing has proven to be a productive and economical method, according to the authors, allowing the description and precise mapping of small regions derived from autosomes, nucleotide polymorphisms, and insertion of LTR retrotransposons, in addition to 2 genes INCENP and SPIRE2, both supposedly involved with the correct segregation of B during cell division.

The DOPseq pipeline [24] has proven to be an elegant strategy for determining chromosomal content and assigning them to genomes, thereby establishing a bridge between cytogenetics and genomics. However, the implementation of this methodology in other vertebrates revealed positive results for analyzing the genetic content of neo-sex chromosomes in the iguanian lizards *Ctenonotus* and *Norops* [37] and the evolutionary dynamics of sex chromosomes in *Anoles* [25]. In addition to the comparative analysis of B microchromosomes in cervids [24, 38, 39], the analysis model was also used to investigate the content of the B chromosome in *Apodemus* field mice [39] and for the first time in the reptilian species *Anolis carolinensis* [35].

To overcome the challenge of sequencing the microdissected DNA from chromosome 1, we employed a secondary approach: analyzing our sequencing data with the DOPseq pipeline. Due to the low quantity of microdissected material for sequencing, it must first undergo an amplification process. In this work, we were able to satisfactorily sequence the DNA of microdissected chromosome 1, generating a reliable library of reads, which were mapped to chromosome 3 of the reference genome, representing the largest chromosome of the karyotype, assigned as chromosome 1. Additionally, mapping reads onto scaffolds not yet anchored in the chromosomes can help complete the assembly of this chromosome.

The description of satellitomes (full catalog of satDNAs) for fish species has been an auxiliary tool for studies of karyotypic evolution [9, 29, 40–46]. For example, the location of satellite DNAs on the B chromosomes in three species of *Psalidodon* confirmed the isochromosome nature for two of them, while for the third, probes showed an asymmetry of its arms [36]. In this study, the satDNAs SatA_mex, SatB_mex, and SatD_mex exhibited symmetrical locations in two regions of chromosome 1, whereas SatC_mex revealed the presence of asymmetric bands on this chromosome. These findings expand the observations obtained from C-banding, which suggested this chromosome was an isochromosome. The application of these satDNAs in other related species may enhance our understanding of the relationships among their chromosome 1, which has already been seen in our results in *P. scabripinnis*. Unfortunately, we did not identify specific satDNAs for chromosome 1. The satDNAs selected as negative controls

did not stain chromosome 1 of the species, thus confirming the specificity of the microdissected material.

To comprehensively investigate the apparent lack of recent chromosomal "tectonism" in chromosome 1 of Characidae, complementary approaches could be employed. Firstly, comparative physical probe mapping, involving the hybridization of specific probes from chromosome 1 of *A. mexicanus* onto chromosomes of other species within the genus *Astyanax* and the broader Characidae family. This comparative mapping strategy offers insights into the conservation or divergence of chromosomal structures across closely related taxa, shedding light on the evolutionary dynamics of chromosome 1 within Characidae. Additionally, *in silico* read mapping of chromosome 1 sequences in these species can provide a high-resolution view of sequence conservation and divergence. By correlating chromosomal evolution with species divergence, ecological niche adaptation, and genetic diversity patterns, a comprehensive understanding of the evolutionary forces shaping chromosome 1 dynamics in Characidae can be achieved.

In summary, employing a multidisciplinary approach encompassing comparative physical probe mapping, in silico read mapping, and genomic analyses holds promise for unraveling the enigmatic patterns of chromosomal evolution in chromosome 1 of Characidae. Such investigations deepen our understanding of the genome architecture within this important fish family.

## Supporting information

**S1 Table. Primers used for amplification of gene fragments present on chromosome 1.**
(DOCX)

**S2 Table. Primers used for amplification of Satellite DNA sequences.** SatA, SatB, SatC, and SatD are present on chromosome 1. The satellite sequences SatE and SatF are outside of chromosome 1 and were used to validate the microdissected chromosome.
(DOCX)

**S3 Table. Similarity with satDNAs described in the literature.**
(DOCX)

**S4 Table. Main scaffolds with reads anchored and not yet assigned to chromosomes.**
(DOCX)

## Acknowledgments

The authors thank Monica Renee Stein for the English editing.

## Author Contributions

**Conceptualization:** Maelin Silva, Jonathan Pena Castro, Edivaldo Herculano Correa de Oliveira, Marcelo Bello Cioffi, Fábio Porto-Foresti, Fausto Foresti, Roberto Ferreira Artoni.

**Data curation:** Maelin Silva, Roberto Ferreira Artoni.

**Formal analysis:** Maelin Silva, Duílio Mazzoni Zerbinato Andrade Silva, Jonathan Pena Castro, Roberto Ferreira Artoni.

**Funding acquisition:** Fausto Foresti, Roberto Ferreira Artoni.

**Investigation:** Maelin Silva, Duílio Mazzoni Zerbinato Andrade Silva, Jonathan Pena Castro, Felipe Faix Barby, Thomas Liehr, Marcelo Bello Cioffi, Roberto Ferreira Artoni.

**Methodology:** Maelin Silva, Duílio Mazzoni Zerbinato Andrade Silva, Jonathan Pena Castro, Alex I. Makunin, Thomas Liehr, Marcelo Bello Cioffi.

**Project administration:** Roberto Ferreira Artoni.

**Resources:** Duílio Mazzoni Zerbinato Andrade Silva, Thomas Liehr, Fausto Foresti, Roberto Ferreira Artoni.

**Software:** Duílio Mazzoni Zerbinato Andrade Silva, Alex I. Makunin, Fausto Foresti.

**Supervision:** Edivaldo Herculano Correa de Oliveira, Fábio Porto-Foresti, Fausto Foresti, Roberto Ferreira Artoni.

**Validation:** Maelin Silva, Duílio Mazzoni Zerbinato Andrade Silva, Jonathan Pena Castro, Alex I. Makunin, Felipe Faix Barby, Edivaldo Herculano Correa de Oliveira, Marcelo Bello Cioffi, Fábio Porto-Foresti, Roberto Ferreira Artoni.

**Visualization:** Maelin Silva, Duílio Mazzoni Zerbinato Andrade Silva, Jonathan Pena Castro, Felipe Faix Barby, Edivaldo Herculano Correa de Oliveira, Thomas Liehr, Marcelo Bello Cioffi.

**Writing – original draft:** Maelin Silva, Duílio Mazzoni Zerbinato Andrade Silva, Jonathan Pena Castro, Roberto Ferreira Artoni.

**Writing – review & editing:** Maelin Silva, Duílio Mazzoni Zerbinato Andrade Silva, Jonathan Pena Castro, Marcelo Bello Cioffi, Fausto Foresti, Roberto Ferreira Artoni.

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
