## [Decision Letter · Decision Letter 0]

8 Oct 2024

PONE-D-24-35999Investigation of Astyanax mexicanus (Characiformes, Characidae) chromosome 1 structure reveals unmapped sequences and suggests conserved evolutionPLOS ONE

Dear Dr. Mazzoni Zerbinato A Silva,

Thank you for submitting your manuscript to PLOS ONE. After careful consideration, we feel that it has merit but does not fully meet PLOS ONE’s publication criteria as it currently stands. Therefore, we invite you to submit a revised version of the manuscript that addresses the points raised during the review process.

We look forward to receiving your revised manuscript.

Kind regards,

Maykon Passos Cristiano, D.Sc.

Academic Editor

PLOS ONE

Journal Requirements:

Additional Editor Comments:

The present study addresses the use of classical and molecular cytogenetics to understand the evolutionary relationship of the fish Astyanax mexicanus and its congeners. The authors attempt to establish a relationship between the conserved chromosome 1 among the species A. mexicanus, A. altiparanae, Psalidodon bifasciatus, and P. scabripinnis. In this study, microdissection of chromosome 1 from A. mexicanus was used to develop a probe, which was then applied to the target species and its congeners.

The manuscript is well written, clear, and objective. The reviewers suggest a minor revision; however, I will recommend a more thorough revision of the manuscript before accepting it for publication. I have some suggestions for improving the text, as well as for the presentation of some figures. Please find my comments below.

**Introduction**

In my opinion, the objective of the study is not clearly stated at the end of the introduction. I suggest that the authors clarify the objective of the study clearer (Page 05).

Page 07, lines 140-153. I think the presentation of Figure 2 is confusing. In the text, it is referred to as Fig. 2a or Fig. 2b, but in Figure 2, there are no "a" or "b" labels, for example. I understand that it refers to SatA and SatB, but for better reader comprehension, it would be important to improve the information in Figure 2.

The organization of Figure 2 is a bit confusing. Please improve it by indicating each image as "A," "B," "C," and so on. The information SatE_mex, for example, could be placed in parentheses below the figure. The authors may consider developing another way to indicate each image in Figure 2.

Page 08, lines 170-171. As this sentence is written, I believe it may create some confusion regarding which figure the authors are referring to. In this regard, I suggest that the authors improve the wording of this sentence.

**Captions**

In all the figure captions, a "tiff" is shown after the figure number. Please check if it is indeed meant to be in the figure caption for the manuscript and remove it if necessary.

Line 117 – “Fig 1.tiff.”

Line 155 - “Fig 2.tiff.”

Line 173 - “Fig 3.tiff.”

**Reviewers**

Please address the suggestions made by the reviewers and note that Reviewer 1 included an attached file with their suggestions.

The authors are welcome to contact me if they feel the need for any clarification or additional information.

Reviewers' comments:

Reviewer's Responses to Questions

**Comments to the Author**

1. Is the manuscript technically sound, and do the data support the conclusions?

Reviewer #1: Yes

Reviewer #2: Yes

2. Has the statistical analysis been performed appropriately and rigorously? 

Reviewer #1: N/A

Reviewer #2: N/A

3. Have the authors made all data underlying the findings in their manuscript fully available?

Reviewer #1: Yes

Reviewer #2: Yes

4. Is the manuscript presented in an intelligible fashion and written in standard English?

Reviewer #1: Yes

Reviewer #2: Yes

5. Review Comments to the Author

Reviewer #1: Dear Dr. Silva,

I have carefully read your manuscript entitled "Investigation of Astyanax mexicanus (Characiformes, Characidae) chromosome 1 structure reveals unmapped sequences and suggests conserved evolution". I believe it contains important information on the structure and location of various DNA sequences of chromosome 1 of A. mexicanus and related species. The study is very well-performed and written in good English, and I therefore suggest this paper to be published in PLOS ONE. However, I have a few technical corrections to the text (please see the attached file).

Reviewer #2: Dear editor,

This study presents classical and molecular cytogenetic patterns Astyanax mexicanus. The papaer "The Investigation of Astyanax mexicanus (Characiformes, Characidae) chromosome 1 structure reveals unmapped sequences and suggests conserved evolution" is a useful and very informative paper, discussing important information about the group's chromosomal Evolution. Some points need to be clarified and revised before the manuscript can be considered for publication.

Dear authors, I point some doubts:

• Where were the specimens collected? Were the specimens collected in a single location? I suggest adding the geographic coordinates.

• How many specimens were collected?

• Were the individuals deposited in an ichthyological collection? If so, I suggest indicating the collection number.

• I suggest review the chromosome classification of pairs 24 and 25 in Figure 1A.

• Were the specimens reviewed by a taxonomist?

6. PLOS authors have the option to publish the peer review history of their article (what does this mean?). If published, this will include your full peer review and any attached files.

Reviewer #1: No

Reviewer #2: No

---

## [Author Response · Author response to Decision Letter 0]

28 Oct 2024

Response to reviewers

We thank the reviewers for the careful proofreading of our manuscript. Below, we address all their concerns, as well as the editor’s suggestions.

Journal Requirements:

Answer: The manuscript has been revised to ensure it meets PLOS ONE's style requirements, including file naming conventions. We have used the style templates provided at the provided links.

Answer: A section has been added in the Materials and Methods to include this information.

Answer: A section has been added in the Materials and Methods to include this information.

Answer: After careful consideration, we have decided to remove the phrase “data not shown” from our manuscript, as the data in question are not a core part of the research being presented in our study.

Additional Editor Comments:

The present study addresses the use of classical and molecular cytogenetics to understand the evolutionary relationship of the fish Astyanax mexicanus and its congeners. The authors attempt to establish a relationship between the conserved chromosome 1 among the species A. mexicanus, A. altiparanae, Psalidodon bifasciatus, and P. scabripinnis. In this study, microdissection of chromosome 1 from A. mexicanus was used to develop a probe, which was then applied to the target species and its congeners.

The manuscript is well written, clear, and objective. The reviewers suggest a minor revision; however, I will recommend a more thorough revision of the manuscript before accepting it for publication. I have some suggestions for improving the text, as well as for the presentation of some figures. Please find my comments below.

Introduction

In my opinion, the objective of the study is not clearly stated at the end of the introduction. I suggest that the authors clarify the objective of the study clearer (Page 05).

Answer: We have carefully reviewed your suggestion and have made the necessary changes to clarify the objective of the study, particularly at the end of the introduction.

Page 07, lines 140-153. I think the presentation of Figure 2 is confusing. In the text, it is referred to as Fig. 2a or Fig. 2b, but in Figure 2, there are no "a" or "b" labels, for example. I understand that it refers to SatA and SatB, but for better reader comprehension, it would be important to improve the information in Figure 2.

The organization of Figure 2 is a bit confusing. Please improve it by indicating each image as "A," "B," "C," and so on. The information SatE_mex, for example, could be placed in parentheses below the figure. The authors may consider developing another way to indicate each image in Figure 2.

Answer: Thank you for your valuable feedback on the presentation and organization of Figure 2. We have addressed the issues as follows:

1.We have added labels "A," "B," "C," etc., to each image in Figure 2 for clarity.

2.The information previously labeled as SatA, SatB, etc., has been placed in parentheses below each image to improve readability and comprehension.

We believe these changes will help enhance the clarity and organization of the figure.

Page 08, lines 170-171. As this sentence is written, I believe it may create some confusion regarding which figure the authors are referring to. In this regard, I suggest that the authors improve the wording of this sentence.

Answer: Thank you for pointing out the potential confusion in the sentence. We have revised the wording for clarity. 

 Captions

In all the figure captions, a "tiff" is shown after the figure number. Please check if it is indeed meant to be in the figure caption for the manuscript and remove it if necessary.

Line 117 – “Fig 1.tiff.”

Line 155 - “Fig 2.tiff.”

Line 173 - “Fig 3.tiff.”

Answer: Thank you, you are correct. We have removed the "tiff" from the captions.

Reviewers

Reviewer #1: Dear Dr. Silva,

I have carefully read your manuscript entitled "Investigation of Astyanax mexicanus (Characiformes, Characidae) chromosome 1 structure reveals unmapped sequences and suggests conserved evolution". I believe it contains important information on the structure and location of various DNA sequences of chromosome 1 of A. mexicanus and related species. The study is very well-performed and written in good English, and I therefore suggest this paper to be published in PLOS ONE. However, I have a few technical corrections to the text (please see the attached file).

Answer: All the suggested corrections in the text have been accepted and implemented.

Reviewer #2: Dear editor,

This study presents classical and molecular cytogenetic patterns Astyanax mexicanus. The papaer "The Investigation of Astyanax mexicanus (Characiformes, Characidae) chromosome 1 structure reveals unmapped sequences and suggests conserved evolution" is a useful and very informative paper, discussing important information about the group's chromosomal Evolution. Some points need to be clarified and revised before the manuscript can be considered for publication.

Dear authors, I point some doubts:

• Where were the specimens collected? Were the specimens collected in a single location? I suggest adding the geographic coordinates.

• How many specimens were collected?

• Were the individuals deposited in an ichthyological collection? If so, I suggest indicating the collection number.

Answer: A section has been added in the Materials and Methods to include this information.

• I suggest review the chromosome classification of pairs 24 and 25 in Figure 1A.

Answer: We reviewed the figure and increased the resolution of the original photo and the resolution of the plate for better observation.

• Were the specimens reviewed by a taxonomist?

Answer: Yes, a section has been added in the Materials and Methods to include information for the study subjects.

---

## [Editor Report · Decision Letter 1]

4 Nov 2024

Investigation of Astyanax mexicanus (Characiformes, Characidae) chromosome 1 structure reveals unmapped sequences and suggests conserved evolution

PONE-D-24-35999R1

Dear Dr. Mazzoni Zerbinato A Silva,

We’re pleased to inform you that your manuscript has been judged scientifically suitable for publication and will be formally accepted for publication once it meets all outstanding technical requirements.

Kind regards,

Maykon Passos Cristiano, D. Sc.

Academic Editor

PLOS ONE
---

## [Editor Report · Acceptance letter]

7 Nov 2024

PONE-D-24-35999R1 

PLOS ONE

Dear Dr. Mazzoni Zerbinato A Silva, 

I'm pleased to inform you that your manuscript has been deemed suitable for publication in PLOS ONE. Congratulations! Your manuscript is now being handed over to our production team.

Kind regards, 

on behalf of

Mr. Maykon Passos Cristiano 

Academic Editor

PLOS ONE